# Identification of SNPs Associated with Grain Quality Traits in Spring Barley Collection Grown in Southeastern Kazakhstan

**Yuliya Genievskaya [1,2], Shyryn Almerekova [1], Saule Abugalieva [1,2], Aigul Abugalieva [3], Kazuhiro Sato [4] and Yerlan Turuspekov [1,2,*]**

[1] Laboratory of Molecular Genetics, Institute of Plant Biology and Biotechnology, Almaty 050040, Kazakhstan; y.genievskaya@ipbb.kz (Y.G.); s.almerekova@ipbb.kz (S.A.); s.abugalieva@ipbb.kz (S.A.)

[2] Faculty of Biology and Biotechnology, Al-Farabi Kazakh National University, Almaty 050040, Kazakhstan

[3] Kazakh Research Institute of Agriculture and Plant Growing, Almalybak 040909, Kazakhstan

[4] Institute of Plant Science and Resources, Okayama University, Kurashiki 710-0046, Japan; kazsato@okayama-u.ac.jp

[*] Correspondence: y.turuspekov@ipbb.kz; Tel.: +7-7273948006

**Abstract:** Barley (*Hordeum vulgare* L.) is an important cereal crop with high genome plasticity that is cultivated in all climatic zones. Traditionally, barley grain is used for animal feed, malting, brewing, and food production. Depending on the end-use product, there are individual requirements for the quality traits of barley grain, particularly for raw starch and protein contents. This study evaluates a collection of 406 two-rowed spring barley accessions, comprising cultivars and lines from the USA, Kazakhstan, Europe, and Africa, based on five grain quality traits (the contents of raw starch, protein, cellulose, and lipids, and grain test weight) over two years. The results of population structure analysis demonstrate the significant impact of geographical origin on the formation of subclusters in the studied population. It was also found that the environment significantly affects grain quality traits. Heat and drought stresses, particularly during grain filling, led to higher protein and lower starch contents. A genome-wide association study (GWAS) using a multiple-locus mixed linear model (MLMM) allowed for the identification of 26 significant quantitative trait loci (QTLs) for the five studied grain quality traits. Among them, 17 QTLs were found to be positioned close to known genes and previously reported QTLs for grain quality in the scientific literature. Most of the identified candidate genes were dehydration stress and flowering genes, confirming that exposure to heat and drought stresses during grain filling may lead to dramatic changes in grain quality traits, including lower starch and higher protein contents. Nine QTLs were presumably novel and could be used for gene mining and breeding activities, including marker-assisted selection to improve grain quality parameters.

**Keywords:** cellulose; grain test weight; GWAS; *Hordeum vulgare* L.; lipids; marker-assisted selection; protein; starch

## 1. Introduction

Barley (*Hordeum vulgare* L.) is the fourth most cultivated cereal crop in the world after corn, wheat, and rice [1], and the second most cultivated cereal crop in Kazakhstan after wheat [2]. Globally, barley cultivation is mostly oriented toward the production of animal feed (about 70%), malting (20–25%), and food (5–10%) [3]. One of the main factors in barley breeding for these purposes is the quality of the grain, including chemical composition and physical properties. The main component of barley grain is carbohydrates, which occupy 78–84% of the total grain, which includes starch (52–72%), β-glucans (4–6%), pentosans (4–8%), and cellulose (1.5–5%) [4]. Besides carbohydrates, barley grain includes proteins (10–17%), free lipids (2–3%), a small percentage of minerals, vitamins (especially vitamin E), dietary fibers, and antioxidants [5]. Requirements for chemical characteristics of high-quality grain depend on the final product. For example, barley grain

used for malting should contain raw protein ranging between 9.5% and 12.5% and raw starch > 60% [6], whereas for feed and food products, higher contents of raw protein and starch are required [7].

The current study focuses on four important biochemical traits and one physical trait of barley grain, namely the contents of raw starch (GSC, %), raw protein (GPC, %), cellulose (GCC, %), and lipids (GLC, %) as biochemical traits and grain test weight per liter (TWL, g/L) as the physical trait. All of the abovementioned characteristics are complex quantitative traits controlled by multi-stage metabolic pathways involving many genetic factors. For example, the biosynthesis of starch in barley grain is mediated by multiple isoforms of starch synthases, starch-branching enzymes, and debranching enzyme isoamylase [8,9]. As for the barley grain proteins, 30–50% of them are hordeins belonging to the prolamin group [10,11]. Major genes involved in the biosynthesis of hordeins are *Hor1* (chromosome 1H), *Hor2* (chromosome 1H), and *Hor5* (chromosome 1H) [12,13]. The remaining barley proteins are albumins, globulins, and glutelin [11]. Previously, it was determined that in barley, two important homologs of the well-studied wheat gene *NAM-B1*—*HvNAM-1* (chromosome 6H) and *HvNAM-2* (chromosome 2H)—are associated with GPC [14,15]. These genes encode transcription factors of the *NAC* family that are linked with accelerating senescence and increasing nutrient remobilization from leaves to developing grains in wheat [16]. The loss of functionality of the *HvNAM-1* gene in barley leads to lower GPC [14]. Nonetheless, it has been shown that *HvNAM-1* and *HvNAM-2* are not highly polymorphic, and their effect on GPC in barley grain is limited [17]. Therefore, GPC is most likely associated with other genetic loci responsible for protein biosynthesis in barley grain. Lipids of barley grain mostly consist of linoleic acid (50.7–57.9% of all lipids) followed by smaller proportions of palmitic (18.3–27.0%), oleic (12.2–21.2%), and linolenic (4.3–7.1%) acids [18]. There are studies suggesting that the *WIN1/SHN1* (chromosome 6H) gene plays an important role in the regulation of lipid biosynthesis pathways in barley grain [19] and the *Nud* gene (chromosome 7H, hulled/hulless grain) probably regulates the composition of lipids in pericarp epidermis [20]. The synthesis of cellulose in plants is regulated by a large cellulose synthase (*CesA*) gene superfamily [21]. However, all of the abovementioned pathways of starch, protein, lipids, and cellulose biosynthesis are complex and controlled by many genes, quantitative trait loci (QTLs), and transcription factors distributed throughout the barley genome. Despite the smaller contribution of one QTL to the manifestation of a trait, the plant genome may contain dozens of these QTLs associated with a trait of interest, resulting in a substantial total contribution to the trait's manifestation.

Two basic approaches are commonly used to identify QTLs in plants—interval mapping (IM) and genome-wide association studies (GWASs) [22]. The first method uses a population of lines generated from a cross between two parent lines. The co-segregation of mapped markers and phenotypic trait alleles helps to identify linked markers. IM has previously been used to map QTLs of some barley grain quality traits, including protein content [23–26], starch content [25,27], acid detergent fiber content [25,28], and grain plumpness and test weight [29,30]. However, the efficiency of IM is limited by the genetic diversity of parental lines used for developing the mapping population and by the small number of recombination events that occur per chromosome per generation [31]. By contrast, GWASs take advantage of many recombination events in natural populations with larger genetic diversity considering haplotype segregation and linkage disequilibrium (LD) [32]. This method is now routinely applied for mapping QTLs of barley yield components [33–36], resistance to biotic [37–39] and abiotic [40,41] stress factors, and for grain quality traits (including the contents of starch and protein [15,42], starch, amylose, amylopectin [43], and arabinoxylan [44]). In addition, GWASs have demonstrated efficiency in identifying important alleles of the candidate genes underlying natural variations in barley, such as *VRS2* [45] and *Ppd-H1* [46]. Thus, GWASs can be applied in a large population to identify markers associated with the trait of interest and provide insight into that trait's genetic architecture.

However, GPC, GSC, GCC, GLC, and TWL are not only under genetic control. Several studies demonstrate the significant impact of the environment and genotype × environment interaction on the manifestation of these traits [47–50]. According to our hypothesis, GPC, GSC, GCC, GLC, and TWL are influenced by genotype and environment. In the current study, our primary goal was to identify genetic markers associated with these traits. The secondary goal was to compare the results of two years and detect possible effects of the environment and/or genotype × environment interaction on studied traits. To achieve these goals, the collection of 406 spring barley accessions was studied for two years in the field of southeastern Kazakhstan. Overall, the marker trait associations (MTAs) identified in this study will be valuable tools for breeding high-quality barley.

## 2. Materials and Methods

### 2.1. Barley Germplasm Collection and Genotyping

A collection of 406 spring two-rowed barley accessions included cultivars and lines from the USA (*n* = 264), Kazakhstan (*n* = 95), Europe (*n* = 37), and Africa (*n* = 10) (Table S1). The American part of the collection was obtained from the US Barley Co-ordinated Agricultural Project (CAP) [51,52] and has previously been analyzed in the various GWAS projects [33,34,53–55]. The US and Kazakhstani parts of the collection were also previously used in Kazakhstan's GWAS on adaptation and yield-related traits [35,36].

The DNA of barley accessions from Kazakhstan was extracted from individual 5-day barley seedlings using a modified cetyltrimethylammonium bromide (CTAB) protocol [56]. The accessions from Kazakhstan were genotyped using the Illumina GoldenGate 9K SNP chip at the TraitGenetics Company (TraitGenetics GmbH, Gatersleben, Germany). Dr. T. Blake provided the US accession genotyping data and seed material (Barley CAP collection). The National Bioresource Project, Barley, Japan, provided seed material and genotyping data on the accessions from Europe and Africa. The SNP genotyping results for barley accessions from Kazakhstan, the USA, Europe, and Africa were merged and filtered according to the minor allele frequency (MAF) and SNP call rate. SNPs with MAF < 0.05 and accessions with SNP with missing data > 0.1 were removed from the data set. In total, 1648 SNPs met all criteria and were selected for further analysis. The SNP positions according to the Illumina iSelect2013 (cM) and Barley 50k iSelect SNP Array (bp) map sets were obtained from the Triticeae toolbox [57].

### 2.2. Field Experiment, Analysis of Grain Quality Traits, and Statistics

The collection was grown in the field of LLP "Kazakh Research Institute of Agriculture and Plant Growing" (KRIAPG, southeastern Kazakhstan) in 2020 and 2021. Phenotypic data are provided in Table S2. Average daily temperature and precipitations at KRIAPG were recorded during periods between key stages of plant growth from sowing to the full maturity of grains (Figure 1).

Each accession was grown in two individual replicated 1 m² plots in a rainfed field with 15 cm spaces between neighboring plots. The replications were evaluated per year in a nearest neighbor randomized complete block design (nn-RCBD) with randomly assigned barley accessions. The accessions were grown under uncontrolled natural conditions without any additional treatment (watering, fertilizers, fungicides, etc.). The field experiment design was standardized for both seasons. After harvesting, the seed material of each accession was collected, cleaned, and sent to the laboratory of grain quality at the LLP "KRIAPG" (Almalybak, Kazakhstan). The grain was studied for five grain quality traits: the grain contents of raw starch (GSC, %), raw protein (GPC, %), raw cellulose (GCC, %), and raw lipids (GLC, %), as well as the grain test weight per liter (TWL, g/L). GSC, GPC, GCC, and GLC were measured using an NIRS DS2500 Grain Analyzer (FOSS, Hillerød, Denmark), which had been calibrated by the manufacturer. TWL was determined according to the guide provided by the Canadian Grain Commission (www.grainscanada.gc.ca/en/ (accessed on 30 January 2023)) and converted into g/L. To gain a better understanding of the relationships between grain quality traits with adaptability- and yield-related traits, the

barley collection was studied in 2020 and 2021 for heading time (HT, days), heading to grain maturity time (HMT, days), vegetation period (VP, days), thousand-kernel weight (TKW, g), and grain yield per m$^2$ (YM2, g/m$^2$). HT is the number of days from seedling emergence (more than 50% of seeds sowed in the plot) to heading (more than 50% of plants in the plot). HMT is the period from heading to full seed maturity (more than 50% of plants in the plot) and VP is the period from seedling emergence to maturity (HT + HMT). The cleaned grains from each individual plot were weighed in g for YM2. TKW was measured as a mass of 1000 random grains in g. Frequency distribution histograms, Pearson correlation analysis, ANOVA, and broad-sense heritability ($h^2$) estimation were performed using the R v4.2.1 statistical platform [58] and RStudio v2022.07.1 software [59].

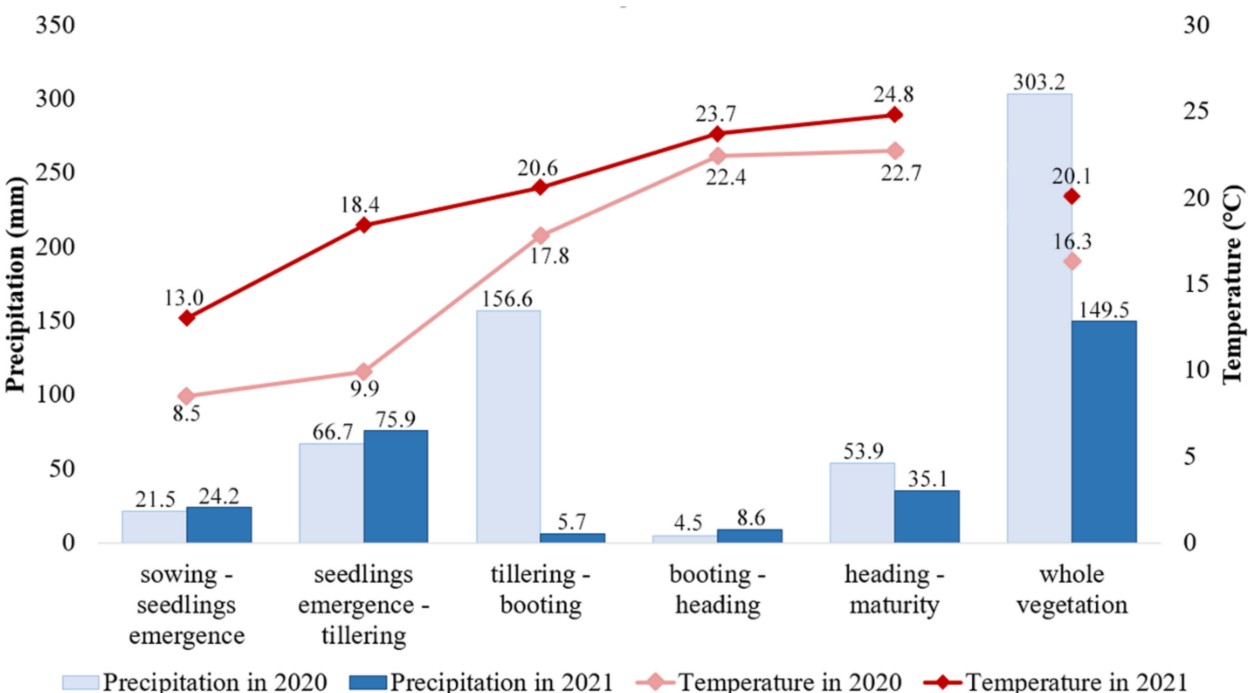

**Figure 1.** Average daily temperature (°C) and precipitation (mm) by plant growth stages at Kazakh Research Institute of Agriculture and Plant Growing (KRIAPG) in 2020 and 2021.

*2.3. Genetic Structure of the Population and the GWAS*

The population structure was determined using three methods: principal component analysis (PCA), neighbor-joining (NJ) clustering method, and clustering with a Bayesian Markov chain Monte Carlo (MCMC) approach based on admixture and correlated allele frequency models (covariance or Q-matrix). PCA was calculated and visualized using the GAPIT v3 package [60] in RStudio v2022.07.1 software. An NJ tree was generated using TASSEL v5.2.84 software [61]. MCMC clustering was performed using STRUCTURE v2.3.4 software [62] with a *K*-value set from 1 to 10, a burn-in period of 100,000, 100,000 MCMC replications after each burn, and an iteration number of 3. The mean *L(K)* and Δ*K* methods [63] of the STRUCTURE HARVESTER v0.6.94 web-based program [64] were used to determine the optimal *K*-value. The Q-matrix was generated based on the optimal *K*-value. To correct for the effects of population substructure in the GWAS, both kinship (K-matrix) and covariance (Q-matrix) were used in the multiple-locus mixed model (MLMM). The MLMM was chosen as one of the most statistically powerful models using forward–backward stepwise linear mixed-model regression to include associated markers as covariates [65]. The GWAS was performed using the GAPIT v3 package for RStudio v2022.07.1. A *p*-value $< 3.14 \times 10^{-5}$ (Bonferroni correction) and a false discovery rate (FDR) < 0.05 were chosen as criteria for significant associations. The linkage disequilibrium (LD) of marker pairs was calculated using TASSEL v5.2.84.

## 3. Results

### 3.1. Genetic Structure of the Barley Population

The population structure of the studied barley collection was analyzed using the three aforementioned methods and accounted for in the GWAS. The PCA results revealed the distinct formation of the US cluster separate from the other accessions as well as the presence of an African cluster, while accessions from Europe and Kazakhstan were clustered together between the USA and Africa (Figure 2A). On average, PC1 accounted for 11.1% of the genetic variation in the population and served to separate the genotypes from the USA, Kazakhstan/Europe, and Africa. In the NJ tree generated from SNP data, three separate subclusters of the US accessions and one subcluster of accessions from Kazakhstan, Europe, and Africa were formed (Figure 2B).

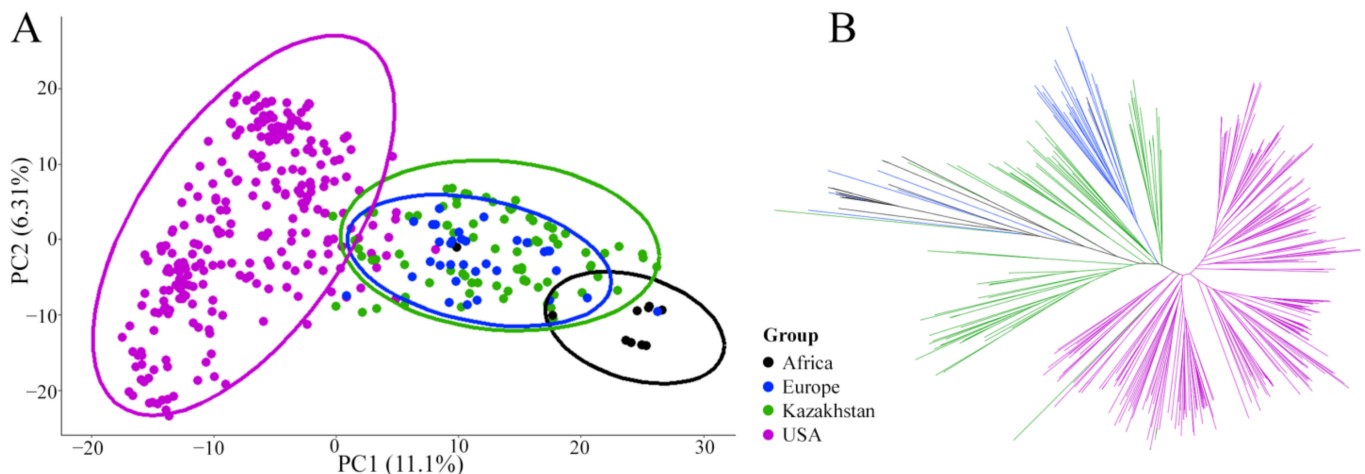

**Figure 2.** Population structure in studied barley collection. (**A**) Principal component analysis (PCA) plot and (**B**) unrooted neighbor-joining (NJ) tree of the studied barley collection including accessions from the USA, Kazakhstan, Europe, and Africa.

The results of STRUCTURE analysis using mean $L(K)$ ($\pm SD$) (Figure 3A) and $\Delta K$ (Figure 3B) methods reveal that the optimal number of subpopulations ($K$) was equal to five, implying the presence of significant population structure in the studied barley collection. The distribution of accessions among subpopulations for $K$ from 2 to 5 is presented in Figure 3C. The composition of subpopulations formed for $K = 5$ is as follows: the Q1 subpopulation—50.7% Kazakhstan, 35.2% Europe, 14.1% Africa, and 0% USA; the Q2 subpopulation—100% Kazakhstan; the Q3 subpopulation —100% USA; the Q4 subpopulation —53.1% USA, 35.4% Kazakhstan, 10.6% Europe, and 0.9% Africa; and the Q5 subpopulation —99.0% USA and 1.0% Kazakhstan. According to the region of origin, accessions under $K = 5$ were distributed among five subpopulations as follows: USA—38.8% in the Q5 subpopulation, 38.4% in the Q3 subpopulation, and 22.8% in the Q4 subpopulation; Kazakhstan—41.7% in the Q4 subpopulation, 37.5% in the Q1 subpopulation, 19.8% in the Q2 subpopulation, and 1.0% in the Q5 subpopulation; Europe—67.6% in the Q1 subpopulation and 32.4% in the Q4 subpopulation; and Africa—90.9% in the Q1 subpopulation and 9.1% in the Q4 subpopulation.

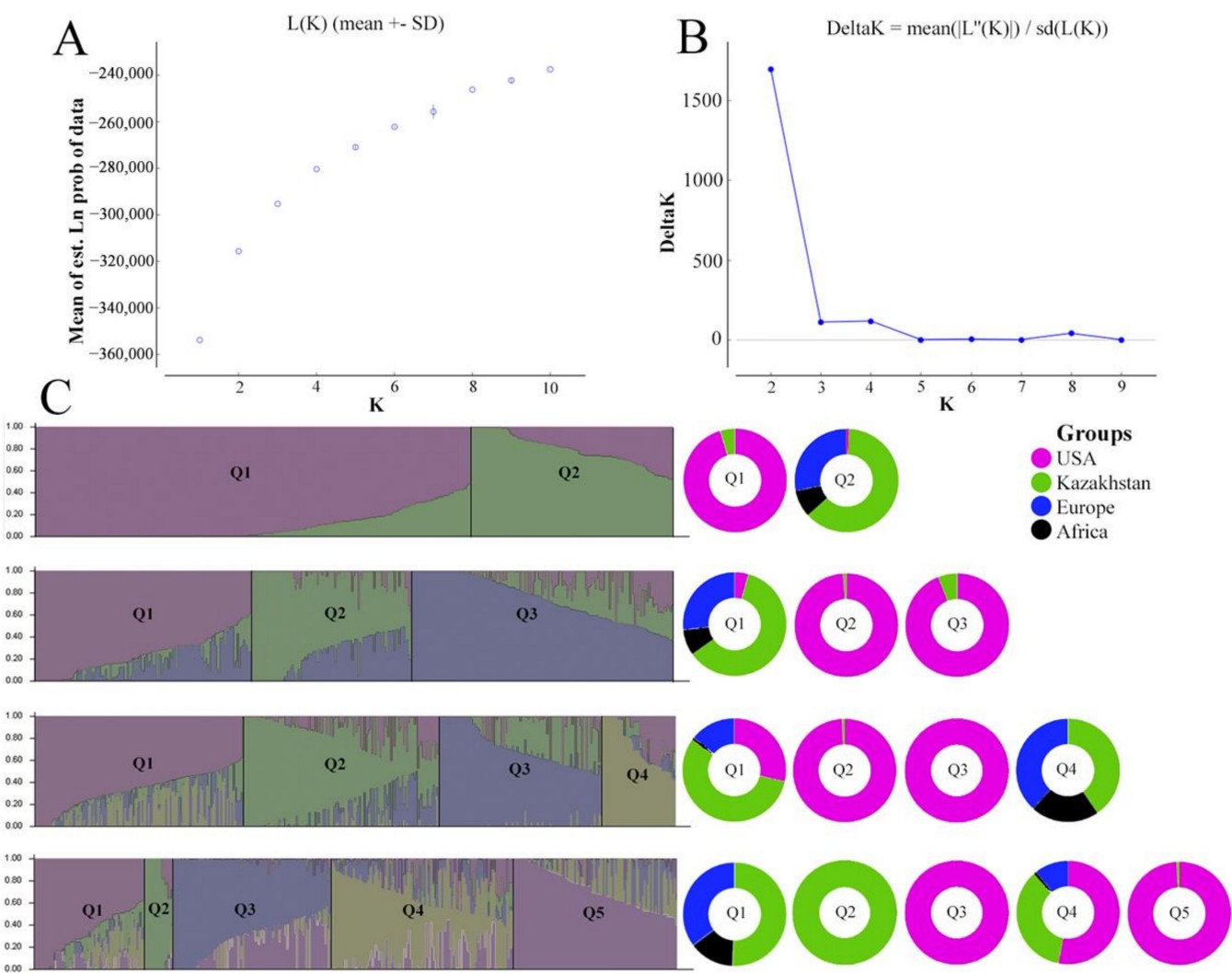

**Figure 3.** Population substructure in the barley collection including accessions from the USA, Kazakhstan, Europe, and Africa. (**A**) Mean *L*(*K*) (±*SD*) for each K value, (**B**) *delta K* (Δ*K*), and (**C**) Bayesian clustering of the 406 barley accessions into groups by *K* from 2 to 5.

### 3.2. Grain Quality Traits

A summary of the information on grain quality traits for two years of experiment as an average of two replications is presented in Table 1. The standard deviations (SDs) for each quality trait across years appear to be stable, indicating that the amount of phenotypic variation in a trait is similar across years, even if the ranges and mean values between years differed (Table 1).

**Table 1.** Summary information for phenotypic data of the five quality traits studied.

| Trait | Year | Range | Median | Mean | SD |
|-------|------|-------|--------|------|-----|
| GSC (%) | 2020 | 50.63–62.80 | 61.48 | 61.14 | 1.33 |
|         | 2021 | 34.86–49.80 | 44.02 | 43.94 | 1.89 |
| GPC (%) | 2020 | 11.65–16.85 | 13.93 | 13.90 | 0.60 |
|         | 2021 | 15.15–22.75 | 18.40 | 18.40 | 1.03 |
| GCC (%) | 2020 | 3.85–6.65 | 5.55 | 5.54 | 0.37 |
|         | 2021 | 3.50–10.46 | 6.00 | 6.17 | 1.04 |

**Table 1.** *Cont.*

| Trait | Year | Range | Median | Mean | SD |
|---|---|---|---|---|---|
| GLC (%) | 2020 | 0.75–3.50 | 2.65 | 2.60 | 0.38 |
|  | 2021 | 0.60–1.95 | 1.40 | 1.37 | 0.18 |
| TWL (g/L) | 2020 | 492.5–688.0 | 582.0 | 581.4 | 31.0 |
|  | 2021 | 419.5–692.0 | 616.0 | 609.7 | 33.6 |

GSC—grain starch content, GPC—grain protein content, GCC—grain cellulose content, GLC—grain lipid content, TWL—grain test weight per liter, SD—standard deviation.

Two-way ANOVA was performed to assess the variance of the five studied grain quality traits with respect to genotype (G), environment or year (E), and genotype × environment interaction (G × E). It was shown that E had a highly significant ($p < 2 \times 10^{-16}$) effect on all studied quality traits (Table 2). The effect of G was significant on GLC ($p = 1.65 \times 10^{-12}$) and highly significant on other traits ($p < 2 \times 10^{-16}$). G×E demonstrated a slightly significant effect on GLC ($p = 0.002$), a significant effect on GSC ($p = 6.26\text{E}{-}14$) and GCC ($p = 5.48 \times 10^{-5}$), and a highly significant effect on GPC and TWL ($p < 2 \times 10^{-16}$) (Table 2). The summary information and the ANOVA results indicate that there is a large amount of grain quality trait variation present in the studied barley collection. The heritability ($h^2$) values were determined for each trait as follows: 0.52 for GCC; 0.34 for TWL; 0.15 for GPC; 0.11 for GLC; and 0.05 for GSC (Table 2).

**Table 2.** ANOVA and heritability of grain quality traits in the studied barley collection.

| GSC | | | | | |
|---|---|---|---|---|---|
|  | df | SS | MS | *p*-Value | $h^2$ |
| G | 406 | 5753 | 14 | $<2 \times 10^{-16}$ | 0.05 |
| E | 1 | 110,551 | 110,551 | $<2 \times 10^{-16}$ | |
| G × E | 387 | 1230 | 3 | $6.26 \times 10^{-14}$ | |
| Res. | 750 | 1258 | 2 | | |
| **GPC** | | | | | |
|  | df | SS | MS | *p*-Value | $h^2$ |
| G | 406 | 1541 | 4 | $<2 \times 10^{-16}$ | 0.15 |
| E | 1 | 7656 | 7656 | $<2 \times 10^{-16}$ | |
| G × E | 387 | 591 | 2 | $<2 \times 10^{-16}$ | |
| Res. | 750 | 377 | 1 | | |
| **GCC** | | | | | |
|  | df | SS | MS | *p*-Value | $h^2$ |
| G | 406 | 1298.4 | 3.2 | $<2 \times 10^{-16}$ | 0.52 |
| E | 1 | 141.8 | 141.78 | $<2 \times 10^{-16}$ | |
| G × E | 387 | 444.2 | 1.15 | $5.48 \times 10^{-5}$ | |
| Res. | 750 | 614.1 | 0.82 | | |
| **GLC** | | | | | |
|  | df | SS | MS | *p*-Value | $h^2$ |
| G | 406 | 88.6 | 0.2 | $1.65 \times 10^{-12}$ | 0.11 |
| E | 1 | 575.2 | 575.2 | $<2 \times 10^{-16}$ | |
| G × E | 387 | 60.4 | 0.2 | 0.00153 | |
| Res. | 750 | 90.4 | 0.1 | | |

**Table 2.** *Cont.*

| | | TWL | | | |
|---|---|---|---|---|---|
| | **df** | **SS** | **MS** | *p*-**Value** | $h^2$ |
| G | 406 | 886,157 | 2183 | $<2 \times 10^{-16}$ | 0.34 |
| E | 1 | 324,351 | 324,351 | $<2 \times 10^{-16}$ | |
| G × E | 387 | 739,572 | 1911 | $<2 \times 10^{-16}$ | |
| Res. | 750 | 634,840 | 846 | | |

G—genotype, E—environment (year), G × E—genotype × environment interaction, Res.—residuals, df—degree of freedom, SS—sum of squares, MS—mean square, $h^2$—broad-sense heritability.

For grain quality traits measured in KRIAPG, correlation coefficients among traits were separately analyzed for two years. In order to determine the strength of the association, absolute values of *r* coefficients were categorized as follows: 0.00–0.19 was regarded as a weak correlation, 0.20–0.59 was regarded as moderate, 0.60–0.89 was regarded as strong, and 0.9–1.00 was regarded as very strong. In 2020, a moderate positive correlation was observed between GCC and GLC (*r* = 0.23), and a weak correlation was observed between GPC and TWL (*r* = 0.15) (Figure 4A). GLC was weakly correlated with GSC and moderately correlated with GPC (*r* = −0.14 and −0.25, respectively). A moderate negative correlation was also observed between GSC and GPC (*r* = −0.34) and between GCC and TWL (*r* = −0.32). In 2021, significant correlations (*p* < 0.05) were observed among all studied traits (Figure 4B). A strong negative correlation was detected between GPC and GSC (*r* = −0.7). A moderate negative correlation was found in pairs GCC/TWL, GSC/GCC, GPC/TWL, and GPC/GLC (*r* ranged from −0.59 to −0.29). GCC was weakly correlated with GLC (*r* = −0.18). Moderate positive correlations were observed in pairs GSC/TWL, GPC/TWL, and GLC/TWL (*r* from 0.52 to 0.26). Finally, GSC demonstrated a weak positive correlation with GLC (*r* = 0.19). Generally, a stable correlation among quality traits in two years was observed in pairs GSC/GPC (negative), GPC/GLC (negative), and GCC/TWL (negative) (Figure 4A,B).

As for correlations of grain quality traits with adaptability, in 2020, moderate negative correlations were observed in pairs GPC/HT and GPC/VP (*r* = −0.42 and −0.33, respectively), while in pairs GSC/HT and GLC/VP, correlations were weak and positive (*r* = 0.18 and 0.11, respectively) (Figure 4C). In 2021, moderate negative correlations were observed in pairs TWL/HT, GSC/HT, and GLC/HT (*r* ranged from −0.39 to −0.2) and weak negative correlations in pairs TWL/VP and GPC/HMT (*r* = −0.16 and −0.12, respectively) (Figure 4D). Several significant correlations between grain quality traits and yield-related traits were also detected. Moderate positive correlations were observed in 2020 in pairs GPC/TKW (*r* = 0.24) and GSC/YM2 (*r* = 0.22), and GCC was positively correlated with YM2 and TKW (*r* = 0.15 and 0.13, respectively) (Figure 4C). Weak negative correlations with TKW and YM2 were identified for TWL (*r* = −0.13 for both) and between GPC and YM2 (*r* = −0.11). In 2021, moderate positive correlations were observed in pairs TWL/YM2, TWL/TKW, GLC/YM2, and GPC/TKW (*r* from 0.33 to 0.24) (Figure 4D). In addition, a weak positive correlation was found between GSC and YM2 (*r* = 0.19). In the same year, GCC was negatively correlated with YM2 and TKW (*r* = −0.23 and −0.16, respectively), and GPC demonstrated a weak negative correlation with YM2 (*r* = −0.11). Stable correlations between grain quality and yield-related traits in two years were found in pairs GPC/TKW (positive), GSC/YM2 (positive), and GPC/YM2 (negative) (Figure 4C,D). No stable correlations between quality traits and adaptability were detected (Figure 4C,D).

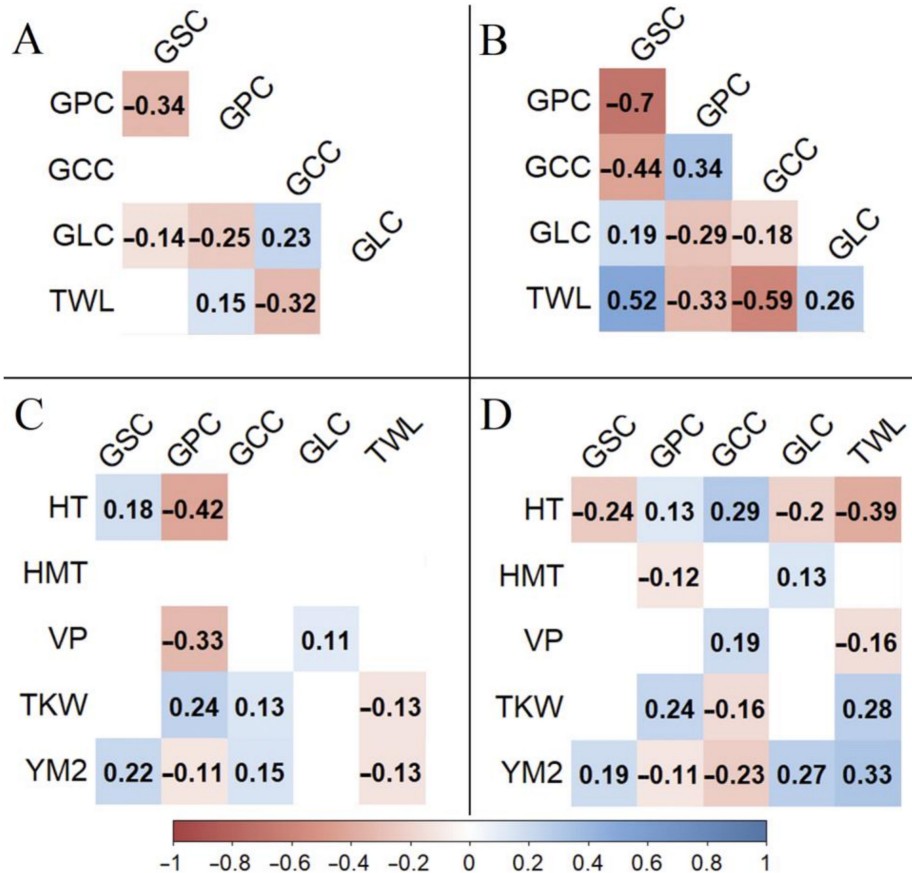

**Figure 4.** Matrices of Pearson correlation coefficients (*r*). The data among grain quality traits in (**A**) 2020 and (**B**) 2021 and between grain quality traits and adaptability- and yield-related traits in (**C**) 2020 and (**D**) 2021. Cells with significance $p < 0.05$ are highlighted in color. The red color denotes a negative correlation and the blue color denotes a positive correlation. Color intensity increases with the decrease in *p*.

### 3.3. Association Analysis and Novel QTLs

A GWAS was separately performed using two-year phenotypic data (2020 and 2021) for each of two repetitions as well as their mean values. Manhattan plots, QQ plots, and full GWAS results are reported in Table S3. Neighboring SNPs associated with the same trait and with R² values (LD) > 0.1 were merged into one QTL. QTLs with *p*-values $< 3.14 \times 10^{-5}$ (threshold of Bonferroni correction) and FDR-adjusted *p*-values < 0.05 in at least one year were selected as significant and are summarized in Table 3. In total, 26 SNPs were identified as being significantly associated with the studied grain quality traits. Identified loci and some important barley genes with known physical positions are shown together on the physical map in Figure 5. For all studied quality traits, there were 16 significant QTLs identified in 2020 and 12 significant QTLs in 2021, with 2 QTLs common to both years (Table 3). The number of significant QTLs by traits is as follows: GSC—8; TWL—8; GPC—5; GLC—3; and GCC—2. QTLs identified for GSC had *p*-values ranging from $1.57 \times 10^{-10}$ to $1.79 \times 10^{-4}$, explaining phenotypic variance from 0.0 to 9.7%. QTLs for TWL demonstrated *p*-values from $2.42 \times 10^{-15}$ to $3.84 \times 10^{-5}$, and phenotypic variance explained (PVE) values from 0.00 to 14.79%. QTLs for GPC had *p*-values ranging from $1.77 \times 10^{-6}$ to $2.75 \times 10^{-5}$ and PVE from 0.00 to 1.44%. For GLC, QTLs demonstrated *p*-values of $1.90 \times 10^{-6}$, $2.93 \times 10^{-5}$, and $5.50 \times 10^{-5}$ and PVE values of 0.54, 1.19, and 0.80%, respectively. The two QTLs for GCC identified in the study had *p*-values of $6.77 \times 10^{-7}$ and $3.10 \times 10^{-8}$ and PVE values of 0.00 and 0.12%, respectively. Boxplots of GSC, GPC, GCC, GLC, and TWL for alleles of major QTLs for each trait are provided in Figure S1.

**Table 3.** Significant marker-trait associations identified in the studied barley collection. *p*-values $< 3.14 \times 10^{-5}$ (Bonferroni criterion) and FDR-adjusted *p*-values $< 0.05$ are highlighted in bold. In the intervals, SNPs with the highest significance are shown. Data for MTAs with *p*-values $> 1.00 \times 10^{-3}$ are not shown.

| Trait | SNP | Chr. | Physical Pos. of SNP (bp) * | QTL Interval (bp) | 2020 | | | | | 2021 | | | | |
|---|---|---|---|---|---|---|---|---|---|---|---|---|---|---|
| | | | | | *p*-Value | *p*-Value (FDR) | PVE (%) | Allele | Effect | *p*-Value | *p*-Value (FDR) | PVE (%) | Allele | Effect |
| GSC | 11_21406 | 2H | 718,210,885 | | | | | | | $8.03 \times 10^{-5}$ | $\mathbf{2.65 \times 10^{-2}}$ | 0.62 | G | 0.41 |
| GSC | 11_20639 | 3H | 158,707,482 | 158,707,482–226,364,211 | $\mathbf{1.57 \times 10^{-10}}$ | $\mathbf{6.66 \times 10^{-7}}$ | 9.70 | A | 1.33 | $\mathbf{8.88 \times 10^{-6}}$ | $\mathbf{3.66 \times 10^{-3}}$ | 0.00 | G | 0.93 |
| GSC | 12_31484 | 3H | 498,949,534 | | $\mathbf{1.43 \times 10^{-6}}$ | $\mathbf{2.96 \times 10^{-3}}$ | 2.90 | A | 0.65 | | | | | |
| GSC | 11_20680 | 4H | 19,087,562 | 19,087,562–20,173,462 | $\mathbf{3.10 \times 10^{-7}}$ | $\mathbf{8.44 \times 10^{-4}}$ | 4.30 | A | 1.12 | | | | | |
| GSC | 11_11473 | 5H | 547,115,792 | | | | | | | $\mathbf{3.12 \times 10^{-6}}$ | $\mathbf{1.72 \times 10^{-3}}$ | 0.26 | C | 0.49 |
| GSC | 11_20104 | 5H | 624,403,396 | 624,403,396–624,444,586 | | | | | | $1.79 \times 10^{-4}$ | $\mathbf{4.91 \times 10^{-2}}$ | 3.90 | G | 0.52 |
| GSC | 12_31042 | 6H | 553,019,586 | 495,778,737–553,203,851 | $5.60 \times 10^{-4}$ | $2.87 \times 10^{-1}$ | 0.39 | G | 0.51 | $\mathbf{5.26 \times 10^{-10}}$ | $\mathbf{8.67 \times 10^{-7}}$ | 0.16 | G | 1.08 |
| GSC | 12_30997 | 7H | 130,414,038 | | $\mathbf{1.35 \times 10^{-5}}$ | $\mathbf{1.11 \times 10^{-2}}$ | 1.44 | A | 0.57 | | | | | |
| GPC | 11_21053 | 1H | 403,309,609 | 403,309,609–481,938,292 | $\mathbf{1.77 \times 10^{-6}}$ | $\mathbf{3.06 \times 10^{-3}}$ | 0.24 | G | 0.46 | $2.74 \times 10^{-4}$ | $2.25 \times 10^{-1}$ | 0.18 | A | 0.59 |
| GPC | 12_20632 | 1H | 511,401,867 | | $\mathbf{2.16 \times 10^{-5}}$ | $\mathbf{1.95 \times 10^{-2}}$ | 0.11 | A | 0.38 | | | | | |
| GPC | 11_20269 | 4H | 72,688,992 | | $\mathbf{9.77 \times 10^{-6}}$ | $\mathbf{1.61 \times 10^{-2}}$ | 0.00 | A | 0.28 | | | | | |
| GPC | 11_21303 | 4H | 464,028,169 | 459,813,388–464,028,169 | $\mathbf{2.75 \times 10^{-5}}$ | $\mathbf{2.42 \times 10^{-2}}$ | 0.00 | G | 0.28 | | | | | |
| GPC | 12_31509 | 6H | 203,509,034 | | | | | | | $\mathbf{5.51 \times 10^{-6}}$ | $\mathbf{9.07 \times 10^{-3}}$ | 1.44 | G | 0.51 |
| GCC | 12_30678 | 2H | UNK | | $\mathbf{3.10 \times 10^{-8}}$ | $\mathbf{5.11 \times 10^{-5}}$ | 0.00 | C | 0.22 | | | | | |
| GCC | 12_11245 | 5H | 579,324,077 | | $\mathbf{6.77 \times 10^{-7}}$ | $\mathbf{1.15 \times 10^{-3}}$ | 0.12 | C | 0.33 | | | | | |
| GLC | 11_21057 | 1H | 478,389,125 | 478,389,125–509,511,424 | $\mathbf{1.90 \times 10^{-6}}$ | $\mathbf{3.14 \times 10^{-3}}$ | 0.54 | G | 0.14 | | | | | |
| GLC | 11_20265 | 5H | 456,062,406 | | $\mathbf{2.93 \times 10^{-5}}$ | $\mathbf{4.83 \times 10^{-2}}$ | 1.19 | A | 0.07 | | | | | |
| GLC | 11_21528 | 7H | 49,445,658 | | $\mathbf{5.50 \times 10^{-5}}$ | $\mathbf{4.53 \times 10^{-2}}$ | 0.80 | T | 0.11 | | | | | |
| TWL | 12_30901 | 2H | 652,031,870 | 652,031,870–705,587,677 | $3.84 \times 10^{-5}$ | $\mathbf{2.11 \times 10^{-2}}$ | 0.18 | G | 9.39 | | | | | |
| TWL | 12_20274 | 4H | 3,623,098 | | | | | | | $\mathbf{5.40 \times 10^{-10}}$ | $\mathbf{8.90 \times 10^{-7}}$ | 14.79 | G | 56.13 |
| TWL | 11_20472 | 4H | 494,212,244 | | | | | | | $\mathbf{4.22 \times 10^{-11}}$ | $\mathbf{1.68 \times 10^{-7}}$ | 0.00 | A | 26.79 |
| TWL | 11_11281 | 5H | 228,224,360 | | | | | | | $\mathbf{9.07 \times 10^{-6}}$ | $\mathbf{1.57 \times 10^{-2}}$ | 0.05 | G | 20.26 |
| TWL | 12_31034 | 5H | 447,605,783 | 397,043,179–447,605,783 | $3.73 \times 10^{-5}$ | $\mathbf{2.11 \times 10^{-2}}$ | 1.18 | C | 9.45 | $\mathbf{8.35 \times 10^{-6}}$ | $\mathbf{6.88 \times 10^{-3}}$ | 0.04 | G | 40.35 |
| TWL | 12_21482 | 6H | 351,737,595 | | | | | | | $\mathbf{3.00 \times 10^{-9}}$ | $\mathbf{5.65 \times 10^{-6}}$ | 0.22 | G | 22.87 |
| TWL | 12_11035 | 7H | 9,613,368 | | $\mathbf{2.42 \times 10^{-15}}$ | $\mathbf{3.99 \times 10^{-12}}$ | 0.25 | G | 24.09 | | | | | |
| TWL | 11_20060 | 7H | 109,656,682 | | | | | | | $\mathbf{3.97 \times 10^{-6}}$ | $\mathbf{3.70 \times 10^{-3}}$ | 0.04 | A | 9.64 |

*—Physical positions according to the Barley 50 K iSelect SNP Array (The Triticeae Toolbox, 2023 [57]), Chr.—chromosome, FDR—false discovery rate, GCC—grain cellulose content, GLC—grain lipids content, GPC—grain protein content, GSC—grain starch content, PVE—phenotypic variance explained by the QTL, TWL—grain test weight.

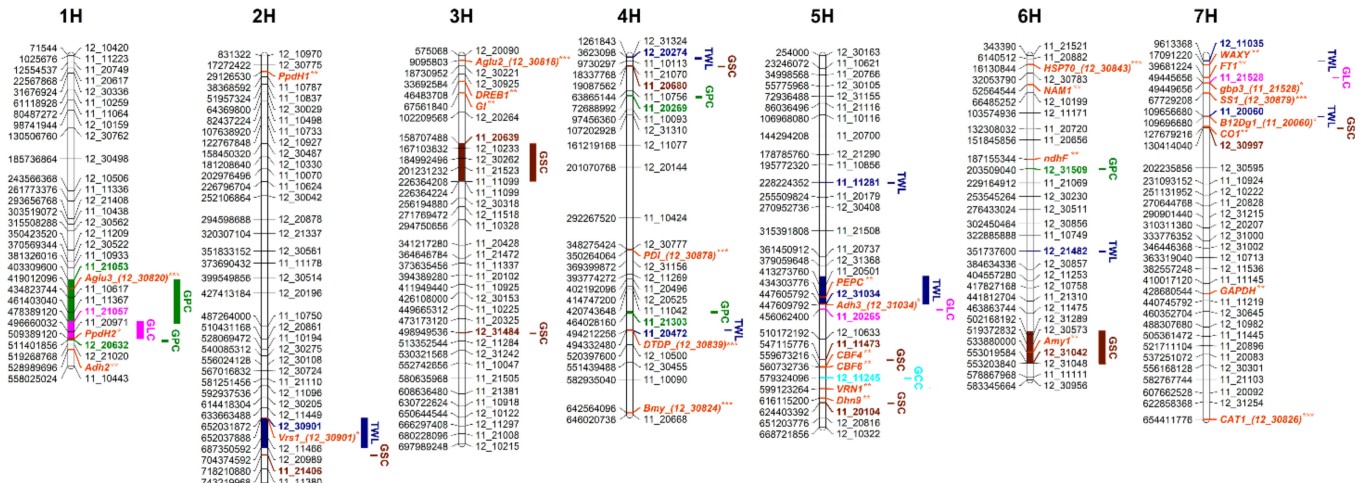

**Figure 5.** Genetic map of QTLs for barley grain quality traits identified in the study. Positions are indicated in base pairs (bps) to the left of the chromosome, with SNP and QTL names indicated on the right. Genes and SNPs associated with traits are highlighted in color. Due to the unknown position of SNP 12_30678, only 25 out of 26 QTLs are shown. *—positions obtained from the Triticeae Toolbox [57]. **—positions obtained from Barley 50k iSelect SNP Array [66]. ***—positions obtained from Szűcs et al. [67].

All QTLs identified in the current study, including their positions in the genome, were compared with QTLs characterized in other studies and with important genes directly or indirectly associated with grain quality in barley. Of the 26 significant QTLs found in the GWAS, the positions of 9 did not match with any grain quality loci from the literature, including 3 QTLs for GSC, 2 QTLs for GPC, 1 QTL for GLC, and 3 QTLs for TWL (Table 4). These QTLs may therefore be considered novel for the studied traits. In the case of the remaining 15 QTLs, they were found to be close to or overlap candidate genes probably associated with these QTLs. For 11 of these QTLs, the literature search revealed candidate QTLs associated with the same grain quality traits.

**Table 4.** Comparison of identified QTLs with candidate genes and QTLs from the literature. Novel QTLs are highlighted in bold.

| Trait | Marker | Chr. | Physical Pos. (bp) * | Genetic Pos. (cM) ** | Key Candidate Genes | Candidate QTLs |
|---|---|---|---|---|---|---|
| **GSC** | **11_21406** | **2H** | **718,210,885** | **143.1** | | |
| GSC | 11_20639 | 3H | 158,707,482–226,364,211 | 58.3–58.4 | | *QTL10_SC* (51.73–55.77 cM) [34]; *qTS-3.1* (176,458,677 bp) [68] |
| **GSC** | **12_31484** | **3H** | **498,949,534** | **-** | | |
| **GSC** | **11_20680** | **4H** | **19,087,562–20,173,462** | **31.1–32.4** | | |
| GSC | 11_11473 | 5H | 547,115,792 | 76.3 | *CBF4* (559,673,235 bp) dehydration-responsive element-binding protein [66]; *CBF5* (560,732,721 bp) dehydration-responsive element-binding protein [66] | *qTS-5.1* (536,435,763 bp) [68] |
| GSC | 11_20104 | 5H | 624,403,396–624,444,586 | 144.8–144.9 | *Dhn9* (616,115,199 bp) dehydrin [66] | |
| GSC | 12_31042 | 6H | 495,778,737–553,203,851 | 73.8–102.0 | *Dhn5* (12_31042, 553,019,586 bp) dehydrin [57]; *Amy1* (533,879,986 bp) alpha-amylase [66] | *QTL18_SC* (71.08 cM) [34] |

**Table 4.** *Cont.*

| Trait | Marker | Chr. | Physical Pos. (bp) * | Genetic Pos. (cM) ** | Key Candidate Genes | Candidate QTLs |
|---|---|---|---|---|---|---|
| GSC | 12_30997 | 7H | 130,414,038 | 74.8 | *CO1* (127,679,215 bp) CONSTANS-like protein [66] | *QTL22_SC* (78.22 cM) [34] |
| GPC | 11_21053 | 1H | 403,309,609–481,938,292 | 51.9–72.9 | *Aglu3* (12_30820, 419,012,101 bp) α-glucosidase [67]; *CO9* (60.0 cM) CONSTANS-like protein [69] | *QTl1_CPC* (55.49 cM) [34] |
| GPC | 12_20632 | 1H | 511,401,867 | - | *Adh2* (528,989,695 bp) alcohol dehydrogenase 2 [66]; *Ppd-H2* (92.3 cM) pseudo-response regulator PPD-H2 [69] | *QTL_Q7* (516,153,706–547,250,913 bp) [70] |
| **GPC** | **11_20269** | **4H** | **72,688,992** | **53.9** | | |
| **GPC** | **11_21303** | **4H** | **459,813,388–464,028,169** | **53.9–54.6** | | |
| GPC | 12_31509 | 6H | 203,509,034 | 58.9 | *ndhF* (187,155,342 bp) nicotinate dehydrogenase FAD-subunit [66] | *QTL_Q24* (12_31509, 203,509,034 bp) [70]; *QGpc6H.45* (54.7 cM) [71]; *Qcp6a* (57.91 cM) [25] |
| GCC | 12_30678 | 2H | UNK | 145.4 | | *QAX2.S-2H4* (136.0 cM) [44] |
| GCC | 12_11245 | 5H | 579,324,077 | 109.4 | *CBF4* (559,673,235 bp) dehydration-responsive element-binding protein [66]; *CBF5* (560,732,721 bp) dehydration-responsive element-binding protein [66] | |
| GLC | 11_21057 | 1H | 478,389,125–509,511,424 | 71.8–90.9 | *Ppd-H2* (92.3 cM) pseudo-response regulator PPD-H2 [69] | |
| **GLC** | **11_20265** | **5H** | **456,062,406** | **44.9** | | |
| GLC | 11_21528 | 7H | 49,445,658 | 49.9 | *FT1* (39,681,222 bp) flowering locus T [66]; *gbp3* (11_21528, 49,445,658 bp) GAMYB-binding protein [57] | |
| TWL | 12_30901 | 2H | 652,031,870–705,587,677 | 90.9–126.6 | *Vrs1* (12_30901, 652,031,870 bp) homeodomain leucine zipper protein [57] | *QTL_Q10* (641,328,117–652,031,870 bp) [70]; *QTw2H.86* (90.99 cM) [71] |
| **TWL** | **12_20274** | **4H** | **3,623,098** | **8.3** | | |
| TWL | 11_20472 | 4H | 494,212,244 | 54.9 | *DTDP* (12_30839, 494,332,468 bp) d-TDP-glucose dehydratase [67] | *QTL_Q14* (11_21303, 464,028,169 bp) [70] |
| **TWL** | **11_11281** | **5H** | **228,224,360** | **45.5** | | |
| TWL | 12_31034 | 5H | 397,043,179–447,605,783 | 44.9–45.0 | *Adh3* (12_31034, 447,605,783 bp) alcohol dehydrogenase 3 [57] | |
| **TWL** | **12_21482** | **6H** | **351,737,595** | **58.9** | | |
| TWL | 12_11035 | 7H | 9,613,368 | 6.3 | *WAXY* (17,091,220 bp) Granule-bound starch synthase 1 [66] | |
| TWL | 11_20060 | 7H | 109,656,682 | 72.8 | *B12Dg1* (11_20060, 109,656,682 bp) B12Dg1 protein [57] | *QTw7H.70* (71.76 cM) [71] |

*—Physical positions according to the Barley 50 K iSelect SNP Array [66], **—Genetic positions according to Illumina iSelect 2013 consensus map [57], Chr.—chromosome, GCC—grain cellulose content, GLC—grain lipids content, GPC—grain protein content, GSC—grain starch content, TWL—grain test weight.

## 4. Discussion

### 4.1. Genetic Structure of the Studied Barley Collection

The population structure in the studied collection may have significantly influenced the GWAS results; therefore, a prior assessment of the analyzed genetic pool is essential in

association mapping [72]. For instance, several studies suggest that growth habit, spike morphology, and geographical origin are primary factors affecting the search for MTAs in diverse barley collections [73–75]. Since the collection in our study was limited to two-rowed spring accessions, the geographical origin is likely one of the primary determinants of the population substructure. On the PCA plot, NJ dendrogram, and STRUCTURE plots for *K* = 2, *K* = 3, and *K* = 4, the US accessions formed separate subclusters with the inclusion of several samples from Kazakhstan (Figures 2 and 3C). The majority of accessions from Kazakhstan and Europe remained unseparated on all population structure plots (Figures 2A,B and 3C) and significantly distinct from the African and US samples (Figure 2A,B). Generally, the results of the NJ analysis confirm that geographical origin affects the substructure in the population, with a few exceptions suggesting admixture among groups (Figures 2B and 3C), which is indicative of a common breeding history with constant germplasm exchange between regions [76,77]. The generated covariance matrix (Q) reflected the genetic differences among origin groups and was applied in the GWAS.

### 4.2. Grain Quality Trait Variation in the Studied Barley Collection

The quality of a GWAS is also strongly influenced by the quality of phenotypic data [78]. In the present study, phenotypic data included five grain quality traits (GSC, GPC, GCC, GLC, and TWL) collected for the GWAS as well as data of three adaptability-related (HT, HMT, and VP) and two yield-related (TKW and YM2) traits additionally collected to gain a better understanding of their effect on grain quality. Data obtained for quality traits in 2020 and 2021 showed adequate ranges and a sufficient amount of phenotypic variation across years (Table 1). Nonetheless, the mean, median, and SD values of the studied traits, as well as the weather conditions (Figure 1), differed between 2020 and 2021 (Table 1). Temperature strongly differed during the period from sowing to seedling emergence (8.5 °C in 2020 and 13.0 °C in 2021) and especially during the period between seedling emergence and tillering (9.9 °C in 2020 and 18.4 °C in 2021). Moreover, in 2021, during the period from tillering to booting, the amount of precipitation was dramatically higher in 2020 than in 2021 (156.6 mm and 5.7 mm, respectively). Generally, during the vegetation period in 2021, the temperature was higher, while the amount of precipitation was lower (Figure 1). The effect of poor water supply and heat stress on spring barley yield and grain quality is well known and well described in the literature [79–81]. The values of yield components and grain quality traits are genetically based, but they can be strongly affected by the moisture and temperature conditions during the vegetation period [82–84]. A corresponding situation was observed in our study, where the effect of E (year) on all studied quality traits prevailed over the effect of G and G × E (Table 2). Unfortunately, it led to small values of $h^2$, especially for GSC, for which $h^2$ = 0.05 (Table 2). This was expected, as starch is a major component of barley grain and is highly dependent on water and temperature conditions [84,85], as well as a high grain protein level, and such effects are usually observed in barley grown under high temperature stress during grain filling [85]. At the same time, GCC is mostly controlled genetically (by cellulose synthase-like *CslF* genes [86]) and is less impacted by heat/drought stress. This is probably why environmental stress had less effect on GCC in the current study ($h^2$ = 0.52) (Table 2).

Variations in GSC, GPC, and other quality traits can be partially explained by heat and drought stress during the grain filling period. The increased temperature usually causes an increase in the rate and a decrease in the grain filling duration [85]. In our study, in 2020, when weather conditions were favorable, longer HT led to higher GSC and lower GPC, while other quality traits in the collection remained unaffected (Figure 4C). On the opposite, in 2021, when heat and drought stress occurred, longer HT was associated with lower GSC, GLC, and TWL, as well as higher GCC and GPC (Figure 4D). HMT was positively correlated with GLC and negatively correlated with GPC in 2021 only (Figure 4D) and did not affect any quality traits in 2020 (Figure 4C). Generally, HT, HMT, and VP were higher in 2020, probably due to more favorable conditions resulting in lower GPC, GCC, and TWL and higher GSC and GLC in the grain (Table 1), which are good for malting barley.

As for the correlations between yield-related and grain quality traits, GPC demonstrated positive correlations with YM2 and TKW in both years, while GSC was positively correlated with YM2 only (Figure 4C,D). Stable negative correlations between quality traits were also observed for the pairs GPC/GSC, GPC/GLC, and GCC/TWL (Figure 4A,B). In other cases, correlations were significant in only one year, and the signs of correlation coefficients were inconsistent.

Despite the effect of E and small values of $h^2$ among the studied grain quality traits, the role of genotype in the estimation of these traits was significant ($p < 2 \times 10^{-16}$) (Table 2), indicating the presence of genes and/or QTLs. Many examples of genes and QTLs for grain quality traits of barley are mentioned in the Introduction. Generally, high phenotypic and genotypic diversity in the studied barley collection provides a solid basis for a robust GWAS.

*4.3. QTLs Associated with Grain Quality Traits in the Studied Barley Collection*

Of all the modern statistical models, MLMM has been described as the best for structured populations in terms of power and false discovery rate [65,87]. Due to the presence of a considerable population structure in our studied collection, we selected MLMM. QQ plots generated in the GWAS demonstrated good fitting to the model with minimal deviation from the line and suggested the minimization of the population structure effect (Table S3).

In this study, 29 out of 71 identified MTAs were selected based on Bonferroni correction ($p < 3.14 \times 10^{-5}$) and FDR ($p < 0.05$) (Table S3). Among these MTAs, some SNPs united within one interval QTL and were considered to be linked to each other (LD $R^2 > 0.01$). This resulted in the identification of 26 QTLs meeting all requirements (Table 3). Only two QTLs were identified in both years, while the other QTLs were found in either 2020 or 2021 (Table 3), confirming the large effect of E (Table 2). Nonetheless, the high significance of these QTLs shows their potential in the manifestation of traits.

Among the studied traits, GSC and GPC were found to be major determinants of the end-use quality of barley grain. There were eight QTLs on chromosomes from 2H to 7H for GSC, explaining 0.0 to 9.7% of phenotypic variance (Table 3). The largest PVE (9.7%) was observed for SNP 11_20639 (chromosome 3H) and GSC (Figure S1). The same genomic region has previously been described as associated with GSC (Table 4). Interestingly, SNPs 11_11473, 11_20104, and 12_31042 associated with GSC were found positioned close to dehydration-associated genes *CBF4*, *CBF5* [88], *Dhn9*, and *Dhn5* [89] (Table 4, Figure 5). All three SNPs were identified in 2021, which was relatively dry and hot, suggesting the dependence of GSC on water and temperature conditions, as described earlier [84,85], and, probably, the involvement of dehydration-associated genes in barley grain quality. Three QTLs for GSC (SNPs 11_21406, 12_31484, and 11_20680) identified in this study did not match any genes or QTLs for GSC described in previous studies (Table 4) and may be considered novel.

For GPC, five QTLs identified on chromosomes 1H, 4H, and 6H explained 0.00 to 1.44% of GPC variance (Table 3, Figure 5). Three of these QTLs were found in 2020, one QTL was found in 2021, and one QTL was found for both years. Three QTLs (SNPs 11_21053, 12_20632, and 12_31509) found close to genes were directly or indirectly involved in barley grain quality and QTLs for GPC were identified in other works (Table 4). Two SNPs—11_21053 and 12_20632 on chromosome 1H—were close to genes *CO9* and *Ppd-H2* (*HvFT3*) (Table 4), determining circadian rhythms in barley [90]. The dependence of GPC on adaptability-related traits, such as HT, HMT, and VP, was also identified in the current study (Figure 4). However, two were novel QTLs for GPC (SNPs 11_20269 and 11_21303) that have never been described in the literature (Table 4).

For GCC, two QTLs explained 0.00 and 0.12% of GCC variance (Table 3, Figure 5). SNP 12_30678 on chromosome 2H was concurrent with the QTL for arabinoxylan content in barley grain, which is hemicellulose (Table 4, Figure S1). The second SNP 12_11245 on chromosome 5H was probably associated with genes *CBF4* and *CBF5*—dehydration-related genes [88]. For GLC, three QTLs explained 0.80 to 1.19% of the phenotypic variation (Table 3, Figure 5). Two QTLs (SNPs 11_21057 and 11_21528) on chromosomes 1H and 7H,

respectively, were found located in the vicinity of the flowering-associated genes *Ppd-H2* (*HvFT3*) and *HvFT1* [90] (Table 4). The third QTL (SNP 11_20265) on chromosome 5H was the novel one (Table 4).

Eight TWL-associated QTLs were detected on five barley chromosomes—2H, 4H, 5H, 6H, and 7H—and explained from 0.00 to 14.79% of TWL variations (Table 3, Figure 5). TWL is a complex trait reflecting barley grain density and is mostly related to GSC as the main grain component. This perhaps explains the fact that two QTLs (SNPs 11_20472 and 12_11035) were positioned close to genes *DTDP* and *WAXY* (Table 4) involved in grain starch metabolism [9,91]. In addition, two QTLs (SNPs 12_31034 and 11_20060) were part of genes *Adh3* and *B12Dg1* (Table 4) previously reported to be involved in the heat- and drought-stress-related tolerance of other cereals [92,93]. One QTL (SNP 12_30901) is a part of the *Vrs1* gene (Table 4), determining the development (suppression) of lateral spikelets [94]. However, the studied collection only included two-rowed barley, and TWL variations may be associated with other functions of *Vrs1*, as previously described in the literature [95]. This study's remaining three QTLs for TWL (SNPs 12_20274, 11_11281, and 12_21482) appear novel (Table 4). Thus, the identified QTLs in this study could potentially be a valuable source for marker-assisted breeding activities, including for the development of kompetitive allele-specific PCR (KASP) arrays [70] for grain quality parameters.

## 5. Conclusions

The studied collection of 406 two-rowed spring barley accessions from the USA, Kazakhstan, Europe, and Africa demonstrated a population structure that was significantly affected by geographical origin. An assessment of the collection using five traits of grain quality (GSC, GPC, GCC, GLC, and TWL) for two years revealed sufficient phenotypic diversity for GWASs. The environment was observed to have a significant role ($p < 2 \times 10^{-16}$) in the manifestation of the studied traits. Heat and drought stress led to higher GPC and lower GSC, particularly during grain filling in 2021 compared with 2020, when the weather conditions were optimal for barley growth. The GWAS using MLMM allowed for the identification of 26 significant QTLs ($p < 3.14 \times 10^{-5}$ and FRD $p < 0.05$) for five studied grain quality traits. Among them, 17 QTLs were found to be close to known genes and QTLs for grain quality, which are reported elsewhere in the literature. The majority of identified candidate genes were found to be associated with dehydration stress and flowering, confirming the negative effect of heat and drought stress, specifically during grain filling. The remaining nine QTLs were presumably novel, as they have not been reported in the scientific literature. Thus, the hypothesis on the effect of the environment and genotype on the quality of barley grain was confirmed. Although the QTLs identified in this study require further detailed research and validation, they could potentially be a valuable source for breeding activities, including KASP technology-based selection for grain quality parameters.

**Supplementary Materials:** The following supporting information can be downloaded at: https://www.mdpi.com/article/10.3390/agronomy13061560/s1. Table S1: The list of accessions used in the study. Table S2: Phenotypic data. Table S3: Full GWAS results, including QQ and Manhattan plots. Figure S1: Boxplots of studied quality traits for alleles of their major QTLs.

**Author Contributions:** Conceptualization, S.A. (Shyryn Almerekova), A.A. and Y.T.; data curation, Y.G., S.A. (Saule Abugalieva) and A.A.; formal analysis, Y.G. and S.A. (Shyryn Almerekova); funding acquisition, S.A. (Shyryn Almerekova); investigation, Y.G. and S.A. (Shyryn Almerekova); methodology, K.S. and Y.T.; project administration, S.A. (Shyryn Almerekova) and Y.T.; resources, K.S., S.A. (Saule Abugalieva) and A.A.; supervision, S.A. (Shyryn Almerekova) and Y.T.; validation, Y.G. and S.A. (Saule Abugalieva); writing—original draft preparation, Y.G. and Y.T.; writing—review and editing, Y.G., S.A. (Shyryn Almerekova), S.A. (Saule Abugalieva), K.S. and Y.T. All authors have read and agreed to the published version of the manuscript.

**Funding:** This research has been funded by the Committee of Science of the Ministry of Science and Higher Education (former Ministry of Education and Science) of the Republic of Kazakhstan (grants nos. AP14871383 and AP08052804).

**Data Availability Statement:** The datasets generated and/or analyzed during the current study are available in the manuscript text and/or Supplementary Materials.

**Acknowledgments:** The authors acknowledge the technical assistance for barley grain quality analysis of the Biochemistry and Grain Quality Laboratory staff at the Kazakh Research Institute of Agriculture and Plant Growing, Almaty region, Kazakhstan.

**Conflicts of Interest:** The authors declare no conflict of interest.

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
