# Peer review of "Identification of SNPs Associated with Grain Quality Traits in Spring Barley Collection Grown in Southeastern Kazakhstan"

_agronomy, doi:10.3390/agronomy13061560_

Round 1

Reviewer 1 Report

The manuscript titled (Identification of SNPs associated with grain quality traits in spring barley collection grown in Southeastern Kazakhstan) by Genievskaya et al. This study evaluates a collection of 406 two-rowed spring barley accessions, comprising cultivars and lines based on five-grain quality traits (contents of raw starch, protein, cellulose, lipids, and grain test weight) over two years. The paper contains interesting results demonstrating a high impact of geographical origin on the studied population. GWAS analysis identified 26 QTLs for the studied traits. The majority of the identified candidate genes are dehydration stress and flowering genes. Nine QTLs are presumably novel and could be used for breeding activities.

The paper is generally well-written and structured. The experiments were successful, and the data was well understood and modeled in detail. In addition, the manuscript contains relevant paragraphs that have been discussed. The selection of the bibliography is appropriate to the content of the manuscript. However, some errors appeared throughout the manuscript, making it difficult to accept it in its current version.

-       Authors should scan the manuscript for minor punctuation and English errors.

-       Arrange the keywords in alphabetical order.

-       The introduction is appropriate, but a few things need further improvements, especially the study hypothesis that should be added for the last five years.

-       Conclusion: Improve this part concerning formulated objectives.

Authors should scan the manuscript for minor punctuation and English errors.

Author Response

Dear Reviewer,

Thank you for all your comments. We are always happy to improve our manuscript. We also considered all your suggestions. The replies to your comments are below:

-       Authors should scan the manuscript for minor punctuation and English errors.

      Thank you for the suggestion. The manuscript has been checked by Professional English proofreading and editing services.

-       Arrange the keywords in alphabetical order.

      Thank you for the note. We rearranged keywords list.

-       The introduction is appropriate, but a few things need further improvements, especially the study hypothesis that should be added for the last five years.

Thank you for the notes. We added hypothesis in the Introduction.

-       Conclusion: Improve this part concerning formulated objectives.

Thank you for the comments. We added several sentences about hypothesis in the Conclusion.

Reviewer 2 Report

Authors should add Boxplots of grain traits in different genotypes at major QTLs.

Author Response

Dear Reviewer,

Thank you for all your comments. We are always happy to improve our manuscript. We also considered your suggestion about boxplots:

Authors should add Boxplots of grain traits in different genotypes at major QTLs.

Thank you for the suggestion. We made 5 boxplots for each trait with their major QTLs and added it as Figure S1.

Reviewer 3 Report

In this study, 406 two-rowed spring barley accessions from various regions were evaluated over a span of two years, focusing on five grain quality traits. The findings revealed that the environment has a significant impact on barley quality. Through the application of genome-wide association study (GWAS), a total of 26 quantitative trait loci (QTLs) were identified, and it was observed that 17 of these QTLs were likely linked to specific genes. The study provides extensive information and raw data in both the main context and supplementary materials. While the text content is detailed, there is room for improvement in presenting the table parts.

Table 1: It is suggested to include "Median" and highlight that the standard deviation (sd) is not small between the two years, such as in the case of GCC.

Table 2: The Anova table can be minimized, retaining only the degree of freedom and p-value, as well as the h2 values.

Table 3: The table format can be modified to resemble Table 1, creating a similar

Author Response

Dear Reviewer,

Thank you for all your comments. We are always happy to improve our manuscript. We also considered all your suggestions. The replies to your comments are below:

Table 1: It is suggested to include "Median" and highlight that the standard deviation (sd) is not small between the two years, such as in the case of GCC.

Thank you for the comment. We added “Median” values in Table 1. Differences in SD values between two years are now mentioned in Discussion together with mean values.

Table 2: The Anova table can be minimized, retaining only the degree of freedom and p-value, as well as the h2 values.

Thank you for the suggestion. We removed F-values from the Table, but we would like to remain SS and MS, since these two values demonstrated absolute amount of variance for each factor indicating which factor is more powerful for the trait.

Table 3: The table format can be modified to resemble Table 1, creating a similar

Thank you for the note. We understand that Table 3 is too big, but it contains only necessary information about QTLs. We think it is also easier for readers to compare QTLs and their effects between two years using this table.